# NEURAL MODEL-BASED REINFORCEMENT LEARNING FOR RECOMMENDATION

## ABSTRACT

There are great interests as well as many challenges in applying reinforcement learning (RL) to recommendation systems. In this setting, an online user is the environment; neither the reward function nor the environment dynamics are clearly defined, making the application of RL challenging. In this paper, we propose a novel model-based reinforcement learning framework for recommendation systems, where we develop a generative adversarial network to imitate user behavior dynamics and learn her reward function. Using this user model as the simulation environment, we develop a novel DQN algorithm to obtain a combinatorial recommendation policy which can handle a large number of candidate items efficiently. In our experiments with real data, we show this generative adversarial user model can better explain user behavior than alternatives, and the RL policy based on this model can lead to a better long-term reward for the user and higher click rate for the system.

## 1 INTRODUCTION

Recommendation systems have become a crucial part of almost all online service platforms. A typical interaction between the system and its users is — users are recommended a page of items and they provide feedback, and then the system recommends a new page of items. A common way of building recommendation systems is to estimate a model which minimizes the discrepancy between the model prediction and the *immediate* user response according to some loss function. In other words, these models do not explicitly take into account the long-term user interest. However, user's interest can evolve over time based on what she observes, and the recommender's action may significantly influence such evolution. In some sense, the recommender is guiding users' interest by displaying particular items and hiding the rest. Thus, a recommendation strategy which takes users' long-term interest into account is more favorable.

Reinforcement learning (RL) is a learning paradigm where a policy will be obtained to guide the actions in an environment so as to maximize the expected long-term reward. Although RL framework has been successfully applied to many game settings, such as Atari (Mnih et al., 2015) and GO (Silver et al., 2016), it met a few challenges in the recommendation system setting because the environment will correspond to the logged online user.

First, a user's interest (reward function) driving her behavior is typically unknown, yet it is critically important for the use of RL algorithms. In existing RL algorithms for recommendation systems, the reward functions are manually designed (e.g. $\pm 1$ for click/no-click) which may not reflect a user's preference over different items (Zhao et al., 2018a; Zheng et al., 2018).

Second, model-free RL typically requires lots of interactions with the environment in order to learn a good policy. This is impractical in the recommendation system setting. An online user will quickly abandon the service if the recommendation looks random and do not meet her interests. Thus, to avoid the large sample complexity of the model-free approach, a model-based RL approach is more preferable. In a related but a different setting where one wants to train a robot policy, recent works showed that model-based RL is much more sample efficient (Nagabandi et al., 2017; Deisenroth et al., 2015; Clavera et al., 2018). The advantage of model-based approaches is that potentially large amount of off-policy data can be pooled and used to learn a good environment dynamics model, whereas model-free approaches can only use expensive on-policy data for learning. However, previous model-based approaches are typically designed based on physics or Gaussian processes, and not tailored for complex sequences of user behaviors.

To address the above challenges, we propose a novel model-based RL framework for recommendation systems, where a user behavior model and the associated reward function are learned in unified minimax framework, and then RL policies are learned using this model. Our main technical innovations are:

1. We develop a generative adversarial learning (GAN) formulation to model user behavior dynamics and recover her reward function. These two components are estimated simultaneously via a joint mini-max optimization algorithm. The benefits of our formulation are: (i) a more predictive user model can be obtained, and the reward function are learned in a consistent way with the user model; (ii) the learned reward allows later reinforcement learning to be carried out in a more principled way, rather than relying on manually designed reward; (ii) the learned user model allows us to perform model-based RL and online adaptation for new users to achieve better results.
2. Using this model as the simulation environment, we also develop a cascading DQN algorithm to obtain a combinatorial recommendation policy. The cascading design of action-value function allows us to find the best subset of items to display from a large pool of candidates with time complexity only linear in the number of candidates.

In our experiments with real data, we showed that this generative adversarial model is a better fit to user behavior in terms of held-out likelihood and click prediction. Based on the learned user model and reward, we show that the estimated recommendation policy leads to better cumulative long-term reward for the user. Furthermore, in the case of model mismatch, our model-based policy can also quickly adapt to the new dynamics with a much fewer number of user interactions compared to model-free approaches.

## 2 RELATED WORK

Commonly used recommendation algorithms typically use a simple user model. For instance, Wide&Deep networks (Cheng et al., 2016) and other methods such as xgboost (Chen & Guestrin, 2016) and DFM (Guo et al., 2017) based on logistic regression essentially assume a user chooses each item independently; Collaborative competitive filtering (Yang et al., 2011) takes into account the context where a user makes her choice but assumes that user's behaviors in each page view are independent. Session-based RNN (Hidasi et al., 2016) and session-based KNN (Jannach & Ludewig, 2017) improve upon previous approaches by modeling users' history, but this model does not recover a users' reward function and can not be used subsequently for reinforcement learning. Bandit based approaches, such as LinUCB (Li et al., 2010), can deal with adversarial user behaviors, but the reward is updated in a Bayesian framework and can not be directly used by a reinforcement learning framework.

Zhao et al. (2018b;a); Zheng et al. (2018) used model-free RL for recommender systems, which may require many user interactions and the reward function is manually designed. Model-based reinforcement learning has been commonly used in robotics applications and resulted in reduced sample complexity to obtain a good policy (Deisenroth et al., 2015; Nagabandi et al., 2017; Clavera et al., 2018). However, these approaches can not be used in the recommendation setting, as a user behavior model typically consists of sequences of discrete choices under a complex session context.

## 3 SETTING AND RL FORMULATION

We will focus on a simple yet typical setting where the recommendation system and its user interact as follows: **a user is displayed to a page of $k$ items and she provides feedback by clicking on one or none of these items, and then the system recommends a new page of $k$ items.** Our model can be extended to settings with more complex page views and user interactions, but these settings are left for future studies.

Since reinforcement learning can take into account long-term reward, it holds the promise to improve users' long-term engagement with an online platform. In the RL framework, a recommendation system wants to find a policy $\pi(s, \mathcal{I})$ to choose a set $\mathcal{I}$ of $k$ items based on user state $s$, such that the long-term expected reward to the user is maximized, i.e.

$$\pi^* = \underset{\pi(s^t, \mathcal{I}^t)}{\arg\max} \; \mathbb{E}\Big[ \sum_{t=0}^{\infty} \gamma^t r(s^t, a^t) \Big], \text{ where } s^0 \sim p^0, \; \mathcal{A}^t \sim \pi(s^t, \mathcal{I}^t), \; s^{t+1} \sim P(\cdot|s^t, \mathcal{A}^t), \; a^t \in \mathcal{A}^t, \quad (1)$$

where several key aspects of this RL framework are as follows:

(1) **Environment**: will correspond to a logged online user who can click on one of the $k$ items displayed by the recommendation system in each page view (or interaction);

(2) **State** $s^t \in \mathcal{S}$: will correspond to an ordered sequence of a user's historical clicks;

(3) **Action** $\mathcal{A}^t \in \binom{\mathcal{I}^t}{k}$ of the recommender: will correspond to a subset of $k$ items chosen by the recommender from $\mathcal{I}^t$ to display to the user. $\binom{\mathcal{I}^t}{k}$ means the set of all subsets of $k$ items of $\mathcal{I}^t$. $\mathcal{I}^t \subset \mathcal{I}$ is the subset of available items to recommend at time $t$ among all items $\mathcal{I}$.

(4) **State Transition** $P(\cdot|s^t, \mathcal{A}^t) : \mathcal{S} \times \binom{\mathcal{I}}{k} \mapsto \mathcal{P}(\mathcal{S})$: will correspond to a user behavior model which returns the transition probability for $s^{t+1}$ given previous state $s^t$ and the set of items $\mathcal{A}^t$ displayed by the system. It is equivalent to the distribution $\phi(s^t, \mathcal{A}^t)$ over a user's actions, which is defined in our user model in section 4.1.

(5) **Reward Function** $r(s^t, \mathcal{A}^t, a^t) : \mathcal{S} \times \binom{\mathcal{I}}{k} \times \mathcal{I} \mapsto \mathbb{R}$: will correspond to a user's utility or satisfaction after making her choice $a^t \in \mathcal{A}^t$ in state $s^t$. Here we assume that the reward to the recommendation system is the same as the user's utility. Thus, a recommendation algorithm which optimizes its long-term reward is designed to satisfy the user in a long run. One can also include the company's benefit to the reward, but in this paper we will focus on users' satisfaction.

(6) **Policy** $\mathcal{A}^t \sim \pi(s^t, \mathcal{I}^t) : \mathcal{S} \times 2^{\mathcal{I}} \mapsto \mathcal{P}(\binom{\mathcal{I}}{k})$: will correspond to a recommendation strategy which takes a user's state $s^t$ and returns the probability of displaying a subset $\mathcal{A}^t$ of $\mathcal{I}^t$.

**Remark.** We note that in the above mapping, *Environment, State* and *State Transition* are associated with the user, the *Action* and *Policy* are associated with the recommendation system, and the *Reward Function* is associated with both the recommendation system and the user. Here we use the notation $r(s^t, \mathcal{A}^t, a^t)$ to emphasize the dependency of the reward on the recommendation action, as the user can only choose from the display set. However, the value of the reward is actually determined by the user's state and the clicked item once the item occurs in the display set $\mathcal{A}^t$. In fact, $r(s^t, \mathcal{A}^t, a^t) = r(s^t, a^t) \cdot \mathbf{1}(a^t \in \mathcal{A}^t)$. Thus, in section 4.1 where we discuss the user model, we simply denote $r(s^t, a^t) = r(s^t, \mathcal{A}^t, a^t)$ and assume $a^t \in \mathcal{A}^t$ is true. The overall RL framework for recommendation is illustrated in Figure 1.

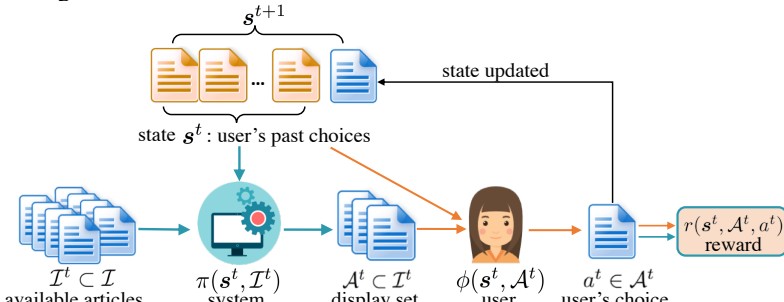

Figure 1: Illustration of the interaction between a user and the recommendation system. Green arrows represent the recommender information flow and orange arrows represent user's information flow.

Since both the reward function and the state transition model are not provided, we need to learn them from data. Once these quantities are learned, the optimal policy $\pi^*$ in Eq. (1) can be estimated by repeated querying to the model using algorithms such as Q-learning (Watkins, 1989). In the next two sections, we will explain our formulation for estimating the user behavior model as well as the reward function and design an efficient algorithm for learning the RL policy for the recommendation.

## 4 GENERATIVE ADVERSARIAL USER MODEL

In this section, we propose a model to imitate users' sequential choices and discuss its parameterization and estimation. The formulation of our user model is inspired by imitation learning, which is a powerful tool for learning sequential decision-making policies from expert demonstrations (Abbeel & Ng, 2004; Ho et al., 2016; Ho & Ermon, 2016; Torabi et al., 2018) We will formulate a unified mini-max optimization to learn user behavior model and reward function simultaneously based on sample trajectories.

### 4.1 USER BEHAVIOR AS REWARD MAXIMIZATION

We model user behavior based on two realistic assumptions. (i) Users are not passive. Instead, when a user is displayed to a set of $k$ items, she will make a choice to maximize her own reward. The reward $r$ measures how much she will be satisfied with or interested in an item. Alternatively, the user can choose not to click on any items. Then she will receive the reward of not wasting time on

boring items. (ii) The reward depends not only on the selected item but also on the user's history. For example, a user may not be interested in *Taylor Swift*'s song at the beginning, but once she happens to listen to it, she may like it and then becomes interested in her other songs. Also, a user can get bored after listening to *Taylor Swift*'s songs repeatedly. In other words, a user's evaluation of the items varies in accordance with her personal experience.

To formalize the model, we consider both the clicked item and the state of the user as the inputs to the reward function $r(\boldsymbol{s}^t, a^t)$, where the clicked item is the user's action $a^t$ and the user's history is captured in her state $\boldsymbol{s}^t$ (non-click is treated as a special item/action). Suppose in session $t$, the user is presented with a set of $k$ items $\mathcal{A}^t = \{a_1, \cdots, a_k\}$ and their associated features $\{\boldsymbol{f}_1^t, \cdots, \boldsymbol{f}_k^t\}$ by the recommendation system. She will take an action $a^t \in \mathcal{A}^t$ according to a strategy $\phi^*$ which can maximize her expected reward. More specially, this strategy is a probability distribution over the set of candidate actions $\mathcal{A}^t$, which is the result of the following optimization problem

$$\text{User Model:} \quad \phi^*(\boldsymbol{s}^t, \mathcal{A}^t) = \arg\max_{\phi \in \Delta^{k-1}} \mathbb{E}_\phi\left[r(\boldsymbol{s}^t, a^t)\right] - R(\phi)/\eta, \tag{2}$$

where $\Delta^{k-1}$ is the probability simplex, and $R(\phi)$ is a convex regularization function to encourage exploration, and $\eta$ controls the strength of the regularization.

**Model Interpretion.** A widely used regularization is the negative Shannon entropy, with which we can obtain an interpretation of our user model from the perspective of exploration-exploitation trade-off (See Appendix A for a proof).

**Lemma 1.** *Let the regularization term in Eq. (2) be $R(\phi) = \sum_{i=1}^k \phi_i \log \phi_i$ and $\phi \in \Delta^{k-1}$ is allowed to be arbitrary mappings. Then the optimal solution $\phi^*$ for the problem in Eq. (2) has a closed form*

$$\phi^*(\boldsymbol{s}^t, \mathcal{A}^t)_i = \exp(\eta r(\boldsymbol{s}^t, a_i)) / \sum_{a_j \in \mathcal{A}^t} \exp(\eta r(\boldsymbol{s}^t, a_j)). \tag{3}$$

*Furthermore, in each session $t$, the user's decision according to her optimal policy $\phi^*$ is equivalent to the following discrete choice model where $\varepsilon^t$ follows a Gumbel distribution.*

$$a^t = \arg\max_{a \in \mathcal{A}^t} \eta\, r(\boldsymbol{s}^t, a) + \varepsilon^t. \tag{4}$$

Essentially, this lemma makes it clear that the user greedily picks an item according to the reward function (exploitation), and yet the Gumbel noise $\varepsilon^t$ allows the user to deviate and explore other less rewarding items. Similar models have also appeared in the econometric choice model (Manski, 1975; McFadden, 1973), but previous econometric models did not take into account diverse features and user state evolution. The regularization parameter $\eta$ is revealed to be an exploration-exploitation trade-off parameter. It can be easily seen that with a smaller $\eta$, the user is more exploratory. Thus, $\eta$ reveals a part of users' character. In practice, we simply set the value $\eta = 1$ in our experiments, since it is implicitly learned in the reward $r$, which is a function of various features of a user.

**Remark.** (i) Other regularization $R(\phi)$ can also be used in our framework, which may induce different user behaviors. In these cases, the relations between $\phi^*$ and $r$ are also different, and may not appear in the closed form. (ii) The case where the user does not click any items can be regarded as a special item which is always in the display set $\mathcal{A}^t$. It can be defined as an item with zero feature vector, or, alternatively, its reward value can be defined as a constant to be learned.

## 4.2 MODEL PARAMETERIZATION

In this section, we will define the state $\boldsymbol{s}^t$ as an embedding of the historical sequence of items clicked by the user before session $t$, and then we will define the reward function $r(\boldsymbol{s}^t, a^t)$ based on the state and the embedding of the current action $a^t$.

First, we will define the state of the user as $\boldsymbol{s}^t := h(\boldsymbol{F}_*^{1:t-1} := [\boldsymbol{f}_*^1, \cdots, \boldsymbol{f}_*^{t-1}])$, where each $\boldsymbol{f}_*^\tau \in \mathbb{R}^d$ is the feature vector of the clicked item at session $\tau$ and $h(\cdot)$ is an embedding function. One can also define a truncated $M$-step sequence as $\boldsymbol{F}_*^{t-m:t-1} := [\boldsymbol{f}_*^{t-m}, \cdots, \boldsymbol{f}_*^{t-1}]$. For the state embedding function $h(\cdot)$, we propose a simple and effective position weighting scheme. Let $\boldsymbol{W} \in \mathbb{R}^{m \times n}$ be a matrix where the number of rows $m$ corresponds to a fixed number of historical steps, and each of the $n$ columns corresponds to one set of importance weights on positions. Then the embedding function $h$ can be designed as

$$\boldsymbol{s}^t = h(\boldsymbol{F}_*^{t-m:t-1}) := vec\left[\sigma\left(\boldsymbol{F}_*^{t-m:t-1}\boldsymbol{W} + \boldsymbol{B}\right)\right] \quad \in \; \mathbb{R}^{dn \times 1}, \tag{5}$$

where $\boldsymbol{B} \in \mathbb{R}^{d \times n}$ is a bias matrix, and $\sigma(\cdot)$ is a nonlinear activation function such as ReLU and ELU, and $vec[\cdot]$ turns the input matrix into a long vector by concatenating the matrix columns.

Alternatively, one can also use an LSTM to capture the history. However, the advantage of the position weighting parameterization is that the history embedding is obtained by a shallow network which is more efficient for forward-computation and gradient backpropagation than RNN.

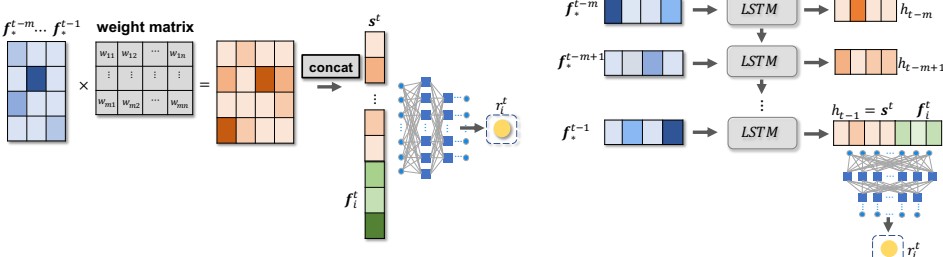

Figure 2: Architecture of our models parameterized by either position weight (PW) or LSTM.

Next, we define the reward function and the user behavior model. A user's choice $a^t \in \mathcal{A}^t$ corresponds to an item with feature $\boldsymbol{f}_{a^t}^t$. Thus we will use $\boldsymbol{f}_{a^t}^t$ as the surrogate for $a^t$ and parameterize the reward function and user behavior model as

$$r(\boldsymbol{s}^t, a^t) := \boldsymbol{v}^\top \sigma \left( \boldsymbol{V} \begin{bmatrix} \boldsymbol{s}^t \\ \boldsymbol{f}_{a^t}^t \end{bmatrix} + \boldsymbol{b} \right) \text{ and } \phi(\boldsymbol{s}, \mathcal{A}^t) \propto \exp \left( \boldsymbol{v}'^\top \sigma \left( \boldsymbol{V}' \begin{bmatrix} \boldsymbol{s}^t \\ \boldsymbol{f}_{a^t}^t \end{bmatrix} + \boldsymbol{b}' \right) \right), \quad (6)$$

where $\boldsymbol{V}, \boldsymbol{V}' \in \mathbb{R}^{\ell \times (dn+d)}$ are weight matrices, $\boldsymbol{b}, \boldsymbol{b}' \in \mathbb{R}^{1 \times (dn+d)}$ are bias vectors , and $\boldsymbol{v}, \boldsymbol{v}' \in \mathbb{R}^\ell$ are the final regression parameters. See Figure 2 for an illustration of the overall parameterization. For simplicity of notation, we will denote the set of all parameters in the reward function as $\theta$ and the set of all parameters in the user model as $\alpha$, and hence the notation $r_\theta$ and $\phi_\alpha$ respectively.

### 4.3 GENERATIVE ADVERSARIAL TRAINING

In practice, both the user reward function $r(\boldsymbol{s}^t, a^t)$ and the behavior model $\phi(\boldsymbol{s}^t, \mathcal{A}^t)$ are unknown and need to be estimated from the data. The behavior model $\phi$ tries to mimic the action sequences provided by a real user who acts to maximize her reward function $r$. In analogy to generative adversarial networks, (i) $\phi$ acts as a generator which generates the user's next action based on her history, and (ii) $r$ acts as a discriminator which tries to differentiate the user's actual actions from those generated by the behavior model $\phi$. Thus, inspired by the GAN framework, we estimate $\phi$ and $r$ simultaneously via a mini-max formulation.

More precisely, given a trajectory of $T$ observed actions $\{a_{true}^1, a_{true}^2, \ldots, a_{true}^T\}$ of a user and the corresponding clicked item features $\{\boldsymbol{f}_*^1, \boldsymbol{f}_*^2, \ldots, \boldsymbol{f}_*^T\}$, we learn the user behavior model and reward function jointly by solving the following mini-max optimization

$$\min_\theta \max_\alpha \left( \mathbb{E}_{\phi_\alpha} \left[ \sum_{t=1}^T r_\theta(\boldsymbol{s}_{true}^t, a^t) \right] - R(\phi_\alpha)/\eta \right) - \sum_{t=1}^T r_\theta(\boldsymbol{s}_{true}^t, a_{true}^t), \quad (7)$$

where we use $\boldsymbol{s}_{true}^t$ to emphasize that this is observed in the data. From the above optimization, one can see that the learned reward function $r_\theta$ will extract some statistics from both real user actions and model user actions, and try to magnify their difference (or make their negative gap larger). In contrast, the learned user behavior model will try to make the difference smaller, and hence more similar to the real user behavior. Alternatively, the mini-max optimization can also be interpreted as a game between an adversary and a learner where the adversary tries to minimize the reward of the learner by adjusting $r_\theta$, while the learner tries to maximize its reward by adjusting $\phi_\alpha$ to counteract the adversarial moves. This gives the user behavior training process a large-margin training flavor, where we want to learn the best model even for the worst scenario.

For general regularization function $R(\phi_\alpha)$, the mini-max optimization problem in Eq. (7) does not have a closed form, and typically needs to be solved by alternatively updating $\phi_\alpha$ and $r_\theta$, e.g.

$$\begin{cases} \alpha \leftarrow \alpha + \gamma_1 \nabla_\alpha \mathbb{E}_{\phi_\alpha} \left[ \sum_{t=1}^T r_\theta(\boldsymbol{s}_{true}^t, a^t) \right] - \gamma_1 \nabla_\alpha R(\phi_\alpha)/\eta; \\ \theta \leftarrow \theta - \gamma_2 \mathbb{E}_{\phi_\alpha} \left[ \sum_{t=1}^T \nabla_\theta r_\theta(\boldsymbol{s}_{true}^t, a^t) \right] + \gamma_2 \sum_{t=1}^T \nabla_\theta r_\theta(\boldsymbol{s}_{true}^t, a_{true}^t). \end{cases} \quad (8)$$

The process may be unstable due to the non-convexity nature of the problem. To stabilize the training process, we will leverage a special regularization for initializing the training process. More specifically, for entropy regularization, we can obtain a closed form solution to the inner-maximization for user behavior model, which makes the learning of reward function easy

(See lemma 2 below and Appendix A for a proof). Once the reward function is learned for entropy regularization, it can be used to initialize the learning in the case of other regularization functions which may induce different user behavior models and final rewards.

**Lemma 2.** *Consider the case where regularization in Eq. (7) is defined as $R(\phi) = \sum_{i=1}^{k} \phi_i \log \phi_i$ and $\Phi$ includes all mappings from $\mathcal{S} \times \binom{\mathcal{I}}{k}$ to $\Delta^{k-1}$. Then the optimization problem in Eq. (7) is equivalent to the following maximum likelihood estimation*

$$\max_{\theta \in \Theta} \prod_{t=1}^{T} \frac{\exp(\eta r_\theta(\boldsymbol{s}_{true}^t, a_{true}^t))}{\sum_{a^t \in \mathcal{A}^t} \exp(\eta r_\theta(\boldsymbol{s}_{true}^t, a^t))}. \tag{9}$$

## 5 CASCADING Q-NETWORKS FOR RL RECOMMENDATION POLICY

Using the estimated user behavior model $\phi$ and the corresponding reward function $r$ as the simulation environment, we can then use reinforcement learning to obtain a recommendation policy. Note that the recommendation policy needs to deal with a *combinatorial action space* $\binom{\mathcal{I}}{k}$, where each action is a subset of $k$ items chosen from a larger set $\mathcal{I}$ of $K$ candidates. Two challenges associated with this problem include the potentially high computational complexity of the combinatorial action space and the development of a framework for estimating the long-term reward (the Q function) from a combination of items. Our contribution is designing a novel cascade of Q-networks to handle the combinatorial action space. We can also design an algorithm to estimate this cascade of Q-networks from interaction with the environment.

### 5.1 CASCADING Q-NETWORKS

We assume that each time when a user visits the online platform, the recommendation system needs to choose a subset $\mathcal{A}$ of $k$ items from $\mathcal{I}$. We will use the Q-learning framework where an optimal action-value function $Q^*(\boldsymbol{s}, \mathcal{A})$ will be learned and satisfies $Q^*(\boldsymbol{s}^t, \mathcal{A}^t) = \mathbb{E}\big[r(\boldsymbol{s}^t, \mathcal{A}^t, a^t) + \gamma \max_{\mathcal{A}' \subset \mathcal{I}} Q^*(\boldsymbol{s}^{t+1}, \mathcal{A}')\big]$, $a^t \in \mathcal{A}^t$. Once the action-value function is learned, an optimal policy for recommendation can be obtained as

$$\pi^*(\boldsymbol{s}^t, \mathcal{I}^t) = \arg\max_{\mathcal{A}^t \subset \mathcal{I}^t} Q^*(\boldsymbol{s}^t, \mathcal{A}^t), \tag{10}$$

where $\mathcal{I}^t \subset \mathcal{I}$ is the set of items available at time $t$. The challenge is that the action space contains $\binom{K}{k}$ many choices, which can be very large even for moderate $K$ (e.g. 1,000) and $k$ (e.g. 5). Furthermore, an item put in different combinations can have different probabilities of being clicked, which is indicated by the user model and is in line with reality. For instance, interesting items may compete with each other for a user's attention. Thus, the policy in Eq. (10) will be very expensive to compute. To address this challenge, we will design not just one but a set of $k$ related Q-functions which will be used in a cascading fashion for finding the maximum in Eq. (10).

Denote the recommender actions as $\mathcal{A} = \{a_1, a_2, \cdots, a_k\} \subset \mathcal{I}$ and the optimal action as $\mathcal{A}^* = \{a_1^*, a_2^*, \cdots, a_k^*\} = \arg\max_{\mathcal{A}} Q^*(\boldsymbol{s}, \mathcal{A})$. Our cascading Q-networks are inspired by the key fact that:

$$\max_{a_1, a_2, \cdots, a_k} Q^*(\boldsymbol{s}, a_1, a_2, \cdots, a_k) = \max_{a_1} \big( \max_{a_2, \ldots, a_k} Q^*(\boldsymbol{s}, a_1, a_2, \cdots, a_k) \big), \tag{11}$$

which also implies that there is a cascade of mutually consistent $Q^{1*}, Q^{2*}, \ldots, Q^{k*}$ such that:

$$a_1^* = \arg\max_{a_1} Q^{1*}(\boldsymbol{s}, a_1) \quad \text{with} \quad Q^{1*}(\boldsymbol{s}, a_1) := \max_{a_2, \cdots, a_k} Q^*(\boldsymbol{s}, a_1, \cdots, a_k),$$

$$a_2^* = \arg\max_{a_2} Q^{2*}(\boldsymbol{s}, a_1^*, a_2) \quad \text{with} \quad Q^{2*}(\boldsymbol{s}, a_1, a_2) := \max_{a_3, \cdots, a_k} Q^*(\boldsymbol{s}, a_1, \cdots, a_k),$$

$$\cdots\cdots$$

$$a_k^* = \arg\max_{a_k} Q^{k*}(\boldsymbol{s}, a_1^*, \cdots, a_{k-1}^*, a_k) \quad \text{with} \quad Q^{k*}(\boldsymbol{s}, a_1, \cdots, a_k) := Q^*(\boldsymbol{s}, a_1, \cdots, a_k).$$

Thus, we can obtain an optimal action in $O(k|\mathcal{I}|)$ computations by applying these functions in a cascading manner. See algorithm 1 and Figure 3 for a summary. However, this cascade of $Q^{j*}$ functions are usually not available and need to be estimated from the data.

### 5.2 PARAMETERIZATION AND ESTIMATION OF CASCADING Q-NETWORKS

Each $Q^{j*}$ function is estimated by a neural network parameterized as

$$\widehat{Q^j}(\boldsymbol{s}, a_{1:j-1}^*, a_j; \Theta_j) = \boldsymbol{q}_j^\top \sigma\Big(\boldsymbol{L}_j \big[\boldsymbol{s}^\top, \boldsymbol{f}_{a_1^*}^\top, \ldots, \boldsymbol{f}_{a_{j-1}^*}^\top, \boldsymbol{f}_{a_j}^\top\big]^\top + \boldsymbol{c}_j\Big), \quad \forall j = 1, \ldots, k, \tag{12}$$

where $\boldsymbol{L}_j \in \mathbb{R}^{\ell \times (dn+dj)}$, $\boldsymbol{c}_j \in \mathbb{R}^\ell$ and $\boldsymbol{q}_j \in \mathbb{R}^\ell$ are the set $\Theta_j$ of parameters, and we use the same embedding for the state $\boldsymbol{s}$ as in Eq. (5). Now the problem left is how we can estimate these functions

$\widehat{Q^j}$. Note that the set of $Q^{j*}$ functions need to satisfy a large set of constraints. At the optimal point, the value of $Q^{j*}$ is the same as $Q^*$ for all $j$, i.e.,

$$Q^{j*}(\boldsymbol{s}, a_1^*, \cdots, a_j^*) = Q^*(\boldsymbol{s}, a_1^*, \cdots, a_k^*), \quad \forall j = 1, \ldots, k. \tag{13}$$

Since it may not be easy to strictly enforce these constraints, we take them into account in a soft and approximate way in our model fitting process as stated below.

Different from standard Q-learning, our cascading Q-learning process is learning a set of $k$ parameterized functions $\widehat{Q^j}(\boldsymbol{s}^t, a_{1:j-1}^*, a_j; \Theta_j)$ as approximations of $Q^{j*}$. To enforce the constraints in Eq. (13) in a soft and approximate way, we can define the loss as

$$\left(y - \widehat{Q^j}\right)^2, \text{ where } y = r(\boldsymbol{s}^t, \mathcal{A}^t, a^t) + \gamma \widehat{Q^k}(\boldsymbol{s}^{t+1}, a_1^*, \cdots, a_k^*; \Theta_k), \ \forall j = 1, \ldots, k. \tag{14}$$

That is all $\widehat{Q^j}$ networks are fitting against the same target $y$. Then the parameters $\Theta_k$ can be updated by performing gradient steps over the above loss. It is noticed in our experiments that the set of learned $\widehat{Q^j}$ networks satisfies the constraints nicely with a small error.

The overall cascading Q-learning algorithm is summarized in Algorithm 2 in Appendix B, where we employ the cascading Q functions to search the optimal action efficiently. Besides, both the experience replay (Mnih et al., 2013) and $\varepsilon$-exploration techniques are applied.

---

**Algorithm 1** Search using $\widehat{Q^j}$ Cascades

1: **function** ARGMAX_Q($\boldsymbol{s}, \mathcal{A}, \Theta_1, \cdots, \Theta_k$)
2:     Let $\mathcal{A}^*$ be empty.
3:     $\mathcal{I} = \mathcal{A} \setminus \boldsymbol{s}$     ▷ remove clicked items.
4:     **for** $j = 1$ to $k$ **do**
5:         $a_j^* = \arg\max_{a_j \in \mathcal{I} \setminus \mathcal{A}^*} \widehat{Q^j}(\boldsymbol{s}, a_{1:j-1}^*, a_j; \Theta_j)$
6:         Update $\mathcal{A}^* = \mathcal{A}^* \cup \{a_j^*\}$
7:     **end for**
8:     **return** $\mathcal{A}^* = (a_1^*, \cdots, a_k^*)$
9: **end function**

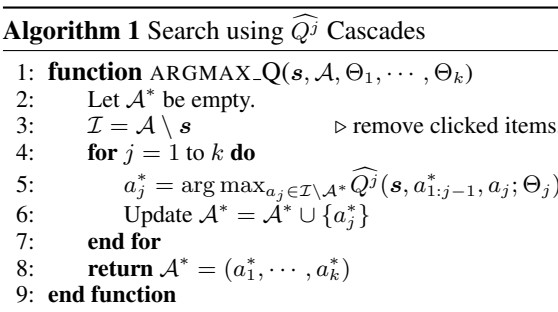
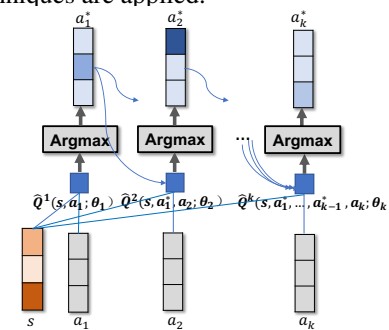

Figure 3: Cascading Q-networks

## 6 Experiments

We conduct three sets of experiments to evaluate our generative adversarial user model (called GAN user model) and the resulting RL recommendation policy. Our experiments are designed to investigate the following questions: **(1)** Can GAN user model lead to better user behavior prediction? **(2)** Can GAN user model lead to higher user reward and click rate? and **(3)** Can GAN user model help reduce the sample complexity of reinforcement learning?

### 6.1 Dataset and feature description

We experimented with 6 real-world datasets: **(1) Ant Financial News dataset** contains clicks records from 50,000 users for one month, involving dozens of thousands of news. On average each display set contains 5 news articles. It also contains user-item cross features which are widely used in this online platform; **(2) MovieLens** contains a large number of movie ratings, from which we randomly sample 1,000 active users. Each display set is simulated by collecting 39 movies released near the time the movie is rated. Movie features are collected from IMDB. Categorical and descriptive features are encoded as sparse and dense vectors respectively; **(3) Last.fm** contains listening records from 359,347 users. Each display set is simulated by collecting 9 songs with the nearest time-stamp. **(4) Yelp** contains users' reviews to various businesses. Each display set is simulated by collecting 9 businesses with the nearest location. **(5) RecSys15** contains click-streams that sometimes end with purchase events. **(6) Taobao** contains the clicking and buying records of users in 22 days. We consider the buying records as positive events. (More details in Appendix C)

### 6.2 Predictive performance of user model

To assess the predictive accuracy of GAN user model with position weight (GAN-PW) and LSTM (GAN-LSTM), we choose a series of most widely used or state-of-the-arts as the baselines, including: (1) W&D-LR (Cheng et al., 2016), a wide & deep model with logistic regression loss function; (2) CCF (Yang et al., 2011), an advanced collaborative filtering model which takes into account the context information in the loss function; we further augment it with wide & deep feature layer (W&D-CCF); (3) IKNN (Hidasi et al., 2015), one of the most popular item-to-item solutions, which calculates items similarly according to the number of co-occurrences in sessions;

(4) S-RNN (Hidasi et al., 2016), a session-based RNN model with a pairwise ranking loss; (5) SCKNNC (Jannach & Ludewig, 2017), a strong methods which unify session based RNN and KNN by cascading combination; (6) XGBOOST (Chen & Guestrin, 2016), a parallel tree boosting; (7) DFM (Guo et al., 2017) is a deep neural factorization-machine based on wide & deep features.

Top-$k$ precision (Prec@$k$) is employed as the evaluation metric. It is the proportion of top-$k$ ranked items at each page view that are actually clicked by the user, averaged across test page views and users. Users are randomly divided into train(50%), validation(12.5%) and test(37.5%) subsets for 3 times. The results are reported in Table 1, which shows that GAN model performs significantly better than baseline models. Moreover, GAN-PW performs nearly as well as GAN-LSTM, but it is more efficient to train. *Thus we use GAN-PW for later experiments and simply refer to it as GAN.*

Table 1: Comparison of predictive performances, where we use Shannon entropy for GAN-PW and GAN-LSTM.

| | (1) Ant Financial news dataset | | (2) MovieLens dataset | | (3) LastFM | |
|---|---|---|---|---|---|---|
| Model | prec(%)@1 | prec(%)@2 | prec(%)@1 | prec(%)@2 | prec(%)@1 | prec(%)@2 |
| IKNN | 20.6(±0.2) | 32.1(±0.2) | 38.8(±1.9) | 40.3(±1.9) | 20.4(±0.6) | 32.5(±1.4) |
| S-RNN | 32.2(±0.9) | 40.3(±0.6) | 39.3(±2.7) | 42.9(±3.6) | 9.4(±1.6) | 17.4(±0.9) |
| SCKNNC | 34.6(±0.7) | 43.2(±0.8) | 49.4(±1.9) | 51.8(±2.3) | 21.4(±0.5) | 26.1(±1.0) |
| XGBOOST | 41.9(±0.1) | 65.4(±0.2) | 66.7(±1.1) | 76.0(±0.9) | 10.2(±2.6) | 19.2(±3.1) |
| DFM | 41.7(±0.1) | 64.2(±0.2) | 63.3(±0.4) | 75.9(±0.3) | 10.5(±0.4) | 20.4(±0.1) |
| W&D-LR | 37.5(±0.2) | 60.9(±0.1) | 61.5(±0.7) | 73.8(±1.2) | 7.6(±2.9) | 16.6(±3.3) |
| W&D-CCF | 37.7(±0.1) | 61.1(±0.1) | 65.7(±0.8) | 75.2(±1.1) | 15.4(±2.4) | 25.7(±2.6) |
| GAN-PW | 41.9(±0.1) | 65.8(±0.1) | 66.6(±0.7) | 75.4(±1.3) | **24.1**(±0.8) | **34.9**(±0.7) |
| GAN-LSTM | **42.1**(±0.2) | **65.9**(±0.2) | **67.4**(±0.5) | **76.3**(±1.2) | 24.0(±0.9) | 34.9(±0.8) |
| | (4) Yelp | | (5) Taobao | | (6) RecSys15: YooChoose | |
| Model | prec(%)@1 | prec(%)@2 | prec(%)@1 | prec(%)@2 | prec(%)@1 | prec(%)@2 |
| IKNN | 57.7(±1.8) | 73.5(±1.8) | 32.8(±2.6) | 46.6(±2.6) | 39.3(±1.5) | 69.8(±2.1) |
| S-RNN | 67.8(±1.4) | 73.2(±0.9) | 32.7(±1.7) | 47.0(±1.4) | 41.8(±1.2) | 69.9(±1.9) |
| SCKNNC | 60.3(±4.5) | 71.6(±1.8) | 35.7(±0.4) | 47.9(±2.1) | 40.8(±2.5) | 70.4(±3.8) |
| XGBOOST | 64.1(±2.1) | 79.6(±2.4) | 30.2(±2.5) | 51.3(±2.6) | 60.8(±0.4) | 80.3(±0.4) |
| DFM | 72.1(±2.1) | 80.3(±2.1) | 30.1(±0.8) | 48.5(±1.1) | **61.3**(±0.3) | **82.5**(±1.5) |
| W&D-LR | 62.7(±0.8) | 86.0(±0.9) | 34.0(±1.1) | 54.6(±1.5) | 51.9(±0.8) | 75.8(±1.5) |
| W&D-CCF | **73.2**(±1.8) | 88.1(±2.2) | 34.9(±1.1) | 53.3(±1.3) | 52.1(±0.5) | 76.3(±1.5) |
| GAN-PW | 72.0(±0.2) | **92.5**(±0.5) | 34.7(±0.6) | 54.1(±0.7) | 52.9(±0.7) | 75.7(±1.4) |
| GAN-LSTM | 73.0(±0.2) | 88.7(±0.4) | **35.9**(±0.6) | **55.0**(±0.7) | 52.7(±0.3) | 75.9(±1.2) |

We also tested different types of regularization (Table 2). In general, Shannon entropy performs well and it is also favored for its closed form solution. However, on the Yelp dataset, we find that $L_2$ regularization $R(\phi) = \|\phi\|_2^2$ leads to a better user model. It is noteworthy that the user model with $L_2$ regularization is trained with Shannon entropy initialization scheme proposed in section 4.3.

Table 2: GAN user model with SE (Shannon entropy) versus $L_2$ regularization on Yelp dataset.

| | Split 1 | | Split 2 | | Split 3 | |
|---|---|---|---|---|---|---|
| Model | prec(%)@1 | prec(%)@2 | prec(%)@1 | prec(%)@2 | prec(%)@1 | prec(%)@2 |
| GAN-LSTM-SE | 73.1 | 88.8 | 72.8 | 89.0 | 73.1 | 88.2 |
| GAN-LSTM-$L_2$ | **73.5** | **89.0** | **78.8** | **91.5** | **76.1** | **91.1** |

Another interesting result on Movielens is shown in Figure 4 (see Appendix D.1 for similar figures). The blue curve represents a user's actual choices over time. The orange curves are trajectories predicted by GAN and W&D-CCF. Each data point $(t, c)$ represents time step $t$ and the category $c$ of the clicked item. The upper sub-figure shows that GAN performs much better as time goes by, while the items predicted by W&D-CCF in the lower sub-figure are concentrated on several categories. This indicates a drawback of static models - it fails to capture the evolution of a user's interests.

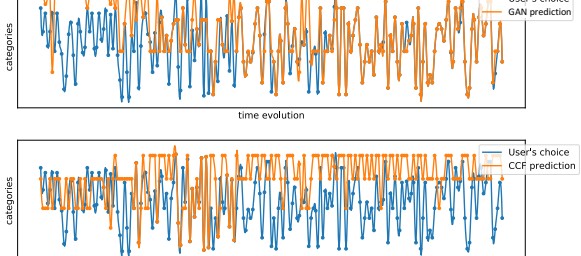

Figure 4: Comparison of the true trajectory (blue) of a user's choices, the simulated trajectory predicted by GAN model (orange curve in upper sub-figure) and the simulated trajectory predicted by W&D-CCF (the orange curve in the lower sub-figure) for the same user. $Y$-axis represents 80 categories of movies.

## 6.3 RECOMMENDATION POLICIES GENERATED FROM USER MODELS

With a learned user model, we can immediately derive a greedy policy to recommend $k$ items with the highest estimated likelihood. We will compare the strongest baseline methods **W&D-LR, W&D-CCF** and **GAN-Greedy** in this setting. Furthermore, we will learn an RL policy using the cascading Q-networks from section 5 (**GAN-CDQN**). We will compare it with two RL methods: a cascading Q-network trained with $\pm 1$ reward (**GAN-RWD1**), and an additive Q-network policy (He et al., 2016), $Q(\boldsymbol{s}, a_1, \cdots, a_k) := \sum_{j=1}^{k} Q(\boldsymbol{s}, a_j)$, trained with the learned reward (**GAN-GDQN**).

Since we cannot perform online experiments at this moment, we use collected data from the online news platform to fit a user model, and then use it as a test environment. To make the experimental results trustful and solid, we fit the test model based on a randomly sampled test set of 1,000 users and keep this set isolated. The RL policies are learned from another set of 2,500 users without overlapping the test set. The performances are evaluated by two metrics: (1) **Cumulative reward**: For each recommendation action, we can observe a user's behavior and compute her reward $r(\boldsymbol{s}^t, a^t)$ using the test model. Note that we never use the reward of test users when we train the RL policy. The numbers shown in Table 3 are the cumulative rewards averaged over time horizon first and then averaged over all users. It can be formulated as $\frac{1}{N} \sum_{u=1}^{N} \frac{1}{T} \sum_{t=1}^{T} r_u^t$, where $r_u^t$ is the reward received by user $u$ at time $t$. (2) **CTR (click through rate)**: it is the ratio of the number of clicks and the number of steps it is run. The values displayed in Table 3 are also averaged over 1,000 test users.

Table 3: Comparison of recommendation performance of different policies.

| model | $k=2$ | | $k=3$ | | $k=5$ | |
|---|---|---|---|---|---|---|
| | reward | CTR | reward | CTR | reward | CTR |
| W&D-LR | 11.82($\pm$0.38) | 0.38($\pm$0.012) | 14.46($\pm$0.42) | 0.46($\pm$0.013) | 15.18($\pm$0.38) | 0.48($\pm$0.011) |
| W&D-CCF | 17.15($\pm$1.16) | 0.53($\pm$0.034) | 19.93($\pm$1.09) | 0.62($\pm$0.031) | 20.94($\pm$1.03) | 0.65($\pm$0.029) |
| GAN-Greedy | 19.17($\pm$1.20) | 0.58($\pm$0.042) | 21.37($\pm$1.24) | 0.67($\pm$0.038) | 22.97($\pm$1.22) | 0.71($\pm$0.034) |
| GAN-RWD1 | 22.37($\pm$0.87) | 0.68($\pm$0.035) | 22.17($\pm$1.07) | 0.68($\pm$0.031) | 25.15($\pm$1.04) | **0.78**($\pm$0.029) |
| GAN-GDQN | 21.88($\pm$0.92) | 0.66($\pm$0.037) | 23.60($\pm$1.06) | 0.72($\pm$0.034) | 23.19($\pm$1.17) | 0.70($\pm$0.033) |
| GAN-CDQN | **22.76**($\pm$0.90) | **0.69**($\pm$0.037) | **24.05**($\pm$0.98) | **0.74**($\pm$0.032) | **25.36**($\pm$1.10) | 0.77($\pm$0.031) |

Three sets of experiments with different numbers of items in each page view are conducted and the results are summarized in Table 3. Since users' behaviors are not deterministic, each policy is evaluated repeatedly for 50 times on test users. The results show that: (1) Greedy policy built on GAN model is significantly better than the policies built on other models. (2) RL policy learned from GAN is better than the greedy policy. (3) Although GAN-CDQN is trained to optimize the cumulative reward, the recommendation policy also achieves a higher CTR compared to GAN-RWD1 which directly optimizes $\pm 1$ reward. The learning of GAN-CDQN may have benefited from the well-known reward shaping effects of the learned continuous reward (Mataric, 1994; Ng et al., 1999; Matignon et al., 2006). (4) While the computational cost of GAN-CDQN is about the same as that of GAN-GDQN (both are linear in the total number of items), our proposed GAN-CDQN is a more flexible parametrization and achieved better results, especially when $k$ is larger.

Since Table 3 only shows average values taken over test users, we compare the policies in user level and the results are shown in figure 5. GAN-CDQN policy results in higher averaged cumulative reward for most users. A similar figure which compares the CTR is deferred to Appendix D. Figure 6 shows that the learned cascading Q-networks satisfy constraints in Eq. (13) well when $k=5$.

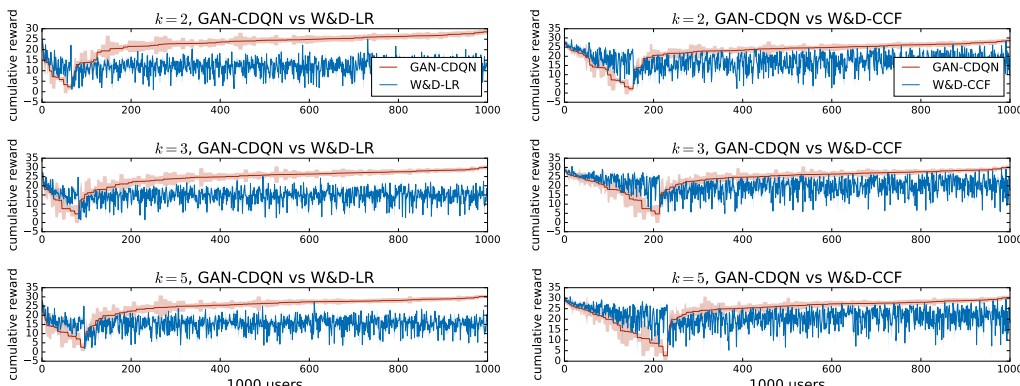

Figure 5: Cumulative rewards among 1,000 users under the recommendation policies based on different user models. The experiments are repeated for 50 times and the standard deviation is plotted as the shaded area.

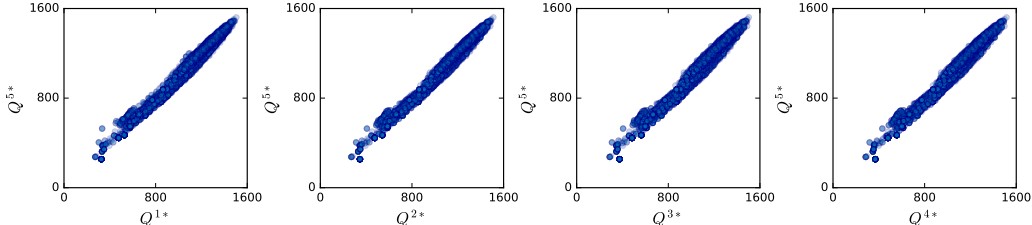

Figure 6: Each scatter-plot compares $Q^{j^*}$ with $Q^{5^*}$ values in Eq. (13) evaluated at the same set of $k$ recommended items. In the ideal case, all scattered points should lie along the diagonal.

## 6.4 USER MODEL ASSISTED POLICY ADAPTATION

Former results in section 6.2 and 6.3 have demonstrated that GAN is a better user model and RL policy based on it can achieve higher CTR compared to other user models, but this user model may be misspecified. In this section, we show that our GAN model can help an RL policy to quickly adapt to a new user. The RL policy assisted by GAN user model is compared with other policies that are learned from and adapted to online users: (1) **CDQN with GAN**: cascading Q-networks which are first trained using the learned GAN user model from other users and then adapted online to a new user using MAML (Finn et al., 2017). (2) **CDQN model free**: cascading Q-networks without pre-trained by the GAN model. It interacts with and adapts to online users directly. (3) **LinUCB**: a classic contextual bandit algorithm which assumes adversarial user behavior. We choose its stronger version - LinUCB with hybrid linear models (Li et al., 2010) - to compare with.

The experiment setting is similar to section 6.3. All policies are evaluated on a set of 1,000 test users associated with a test model. Three sets of results corresponding to different sizes of display set are plotted in Figure 7. It shows how the CTR increases as each policy interacts with and adapts to users over time. In fact, the performances of users' cumulative reward according to different policies are also similar, and the corresponding figure is deferred to Appendix D.3.

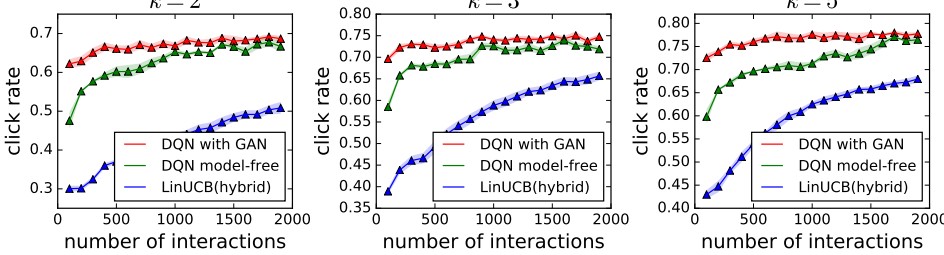

Figure 7: Comparison of the averaged click rate averaged over 1,000 users under different recommendation policies. $X$-axis represents how many times the recommender interacts with online users. $Y$-axis is the click rate. Each point $(x, y)$ means the click rate $y$ is achieved after $x$ times of user interactions.

It can be easily seen that the CDQN policy pre-trained over a GAN user model can quickly achieve a high CTR even when it is applied to a new set of users (Figure 7). Without the user model, CDQN can also adapt to the users during its interaction with them. However, it takes around 1,000 iterations (i.e., 100,000 interactive data points) to achieve similar performance as the CDQN policy assisted by GAN user model. LinUCB(hybrid) is also capturing users' interests during its interaction with users. Similarly, it takes too many interactions. In Appendix D.3, another figure is attached to compare the cumulative reward received by the user instead of CTR. Generally speaking, GAN user model provides a dynamical environment for RL policies to interact with. It helps the policy achieve a more satisfying status before applying to online users.

## 7 CONCLUSION AND FUTURE WORK

We proposed a novel model-based reinforcement learning framework for recommendation systems, where we developed a GAN formulation to model user behavior dynamics and her associated reward function. Using this user model as the simulation environment, we develop a novel cascading Q-network for combinatorial recommendation policy which can handle a large number of candidate items efficiently. Although the experiments show clear benefits of our method in an offline and realistic simulation setting, even stronger results could be obtained via future online A/B testing.

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

## A   LEMMA

### A.1   PROOF OF LEMMA 1

**Lemma 1.** *Let the regularization term in Eq. (2) be $R(\phi) = \sum_{i=1}^{k} \phi_i \log \phi_i$ and $\phi \in \Delta^{k-1}$ is allowed to be arbitrary mappings. Then the optimal solution $\phi^*$ for the problem in Eq. (2) has a closed form*

$$\phi^*(\boldsymbol{s}^t, \mathcal{A}^t)_i = \exp(\eta r(\boldsymbol{s}^t, a_i))/\sum_{a_j \in \mathcal{A}^t} \exp(\eta r(\boldsymbol{s}^t, a_j)). \tag{3}$$

*Furthermore, in each session t, the user's decision according to her optimal policy $\phi^*$ is equivalent to the following discrete choice model where $\varepsilon^t$ follows a Gumbel distribution.*

$$a^t = \arg\max_{a \in \mathcal{A}^t} \eta\, r(\boldsymbol{s}^t, a) + \varepsilon^t. \tag{4}$$

*Proof.* First, recall the problem defined in Eq. (2):

$$\phi^*(\boldsymbol{s}^t, \mathcal{A}^t) = \arg\max_{\phi \in \Delta^{k-1}} \mathbb{E}_\phi\left[r(\boldsymbol{s}^t, a^t)\right] - \frac{1}{\eta} R(\phi).$$

Denote $\phi^t = \phi(\boldsymbol{s}^t, \mathcal{A}^t)$. Since $\phi$ can be an arbitrary mapping (i.e., $\phi$ is not limited in a specific parameter space), $\phi^t$ can be an arbitrary vector in $\Delta^{k-1}$. Recall the notation $\mathcal{A}^t = \{a_1, \cdots, a_k\}$. Then the expectation taken over random variable $a^t \in \mathcal{A}^t$ can be written as

$$\mathbb{E}_\phi\left[r(\boldsymbol{s}^t, a^t)\right] - \frac{1}{\eta} R(\phi) = \sum_{i=1}^{k} \phi_i^t r(\boldsymbol{s}^t, a_i) - \frac{1}{\eta} \sum_{i=1}^{k} \phi_i^t \log \phi_i^t. \tag{15}$$

By simple computation, the optimal vector $\phi^{t*} \in \Delta^{k-1}$ which maximizes Eq. (15) is

$$\phi_i^{t*} = \frac{\exp(\eta r(\boldsymbol{s}^t, a_i))}{\sum_{j=1}^{k} \exp(\eta r(\boldsymbol{s}^t, a_j))}, \tag{16}$$

which is equivalent to Eq. (2). Next, we show the equivalence of Eq. (16) to the discrete choice model interpreted by Eq. (4).

The cumulative distribution function for the Gumbel distribution is $F(\varepsilon; \alpha) = \mathbb{P}[\varepsilon \leqslant \alpha] = e^{-e^{-\alpha}}$ and the probability density is $f(\varepsilon) = e^{-e^{-\varepsilon}} e^{-\varepsilon}$. Using the definition of the Gumbel distribution, the probability of the event $[a^t = a_i]$ where $a^t$ is defined in Eq. (4) is

$$P_i := \mathbb{P}\left[a^t = a_i\right] = \mathbb{P}\left[\eta r(\boldsymbol{s}^t, a_i) + \varepsilon_i \geqslant \eta r(\boldsymbol{s}^t, a_j) + \varepsilon_j, \text{ for all } i \neq j\right]$$

$$= \mathbb{P}\left[\varepsilon_j \leqslant \varepsilon_i + \eta r(\boldsymbol{s}^t, a_i) - \eta r(\boldsymbol{s}^t, a_j), \text{ for all } i \neq j\right].$$

Suppose we know the random variable $\varepsilon_i$. Then we can compute the choice probability $P_i$ conditioned on this information. Let $B_{ij} = \varepsilon_i + \eta r(\boldsymbol{s}^t, a_i) - \eta r(\boldsymbol{s}^t, a_j)$ and $P_{i|\mathcal{E}}$ be the conditional probability; then we have

$$P_{i|\varepsilon_i} = \prod_{i \neq j} \mathbb{P}[\varepsilon_j \leqslant B_{ij}] = \prod_{i \neq j} e^{-e^{-B_{ij}}}.$$

In fact, we only know the density of $\varepsilon_i$. Hence, using the Bayes theorem, we can express $P_i$ as

$$P_i = \int_{-\infty}^{\infty} P_{i|\varepsilon_i} f(\varepsilon_i) \mathrm{d}\varepsilon_i = \int_{-\infty}^{\infty} \prod_{i \neq j} e^{-e^{-B_{ij}}} f(\varepsilon_i) \mathrm{d}\varepsilon_i$$

$$= \int_{-\infty}^{\infty} \prod_{j=1}^{k} e^{-e^{-B_{ij}}} e^{e^{-\varepsilon_i}} e^{-e^{-\varepsilon_i}} e^{-\varepsilon_i} \mathrm{d}\varepsilon_i = \int_{-\infty}^{\infty} \left(\prod_{j=1}^{k} e^{-e^{-B_{ij}}}\right) e^{-\varepsilon_i} \mathrm{d}\varepsilon_i$$

Now, let us look at the product itself.

$$\prod_{j=1}^{k} e^{-e^{-B_{ij}}} = \exp\Big(-\sum_{j=1}^{k} e^{-B_{ij}}\Big)$$

$$= \exp\Big(-e^{-\varepsilon_i}\sum_{j=1}^{k} e^{-(\eta r(\boldsymbol{s}^t,a_i)-\eta r(\boldsymbol{s}^t,a_j))}\Big)$$

Hence

$$P_i = \int_{-\infty}^{\infty} \exp(-e^{-\varepsilon_i}Q)e^{-\varepsilon_i}\mathrm{d}\varepsilon_i$$

where $Q = \sum_{j=1}^{k} e^{-(\eta r(\boldsymbol{s}^t,a_i)-\eta r(\boldsymbol{s}^t,a_j))} = Z/\exp(\eta r(\boldsymbol{s}^t,a_i))$.

Next, we make a change of variable $y = e^{-\varepsilon_i}$. The Jacobian of the inverse transform is $J = \frac{\mathrm{d}\varepsilon_i}{\mathrm{d}y} = -\frac{1}{y}$. Since $y > 0$, the absolute of Jacobian is $|J| = \frac{1}{y}$. Therefore,

$$P_i = \int_0^{\infty} \exp(-Qy)y|J|\mathrm{d}y = \int_0^{\infty} \exp(-Qy)\mathrm{d}y$$

$$= \frac{1}{Q} = \frac{1}{\exp(-\eta r(\boldsymbol{s}^t,a_i))\sum_j \exp(\eta r(\boldsymbol{s}^t,a_j))}$$

$$= \frac{\exp(\eta r(\boldsymbol{s}^t,a_i)}{\sum_{j=1}^{k} \exp(\eta r(\boldsymbol{s}^t,a_j))}.$$

$\square$

## A.2 PROOF OF LEMMA 2

**Lemma 2.** *Consider the case where regularization in Eq. (7) is defined as $R(\phi) = \sum_{i=1}^{k} \phi_i \log \phi_i$ and $\Phi$ includes all mappings from $\mathcal{S} \times \binom{\mathcal{I}}{k}$ to $\Delta^{k-1}$. Then the optimization problem in Eq. (7) is equivalent to the following maximum likelihood estimation*

$$\max_{\theta \in \Theta} \prod_{t=1}^{T} \frac{\exp(\eta r_\theta(\boldsymbol{s}^t_{true}, a^t_{true}))}{\sum_{a^t \in \mathcal{A}^t} \exp(\eta r_\theta(\boldsymbol{s}^t_{true}, a^t))}. \tag{9}$$

*Proof.* This lemma is a straight forward result of lemma 1. First, recall the problem defined in Eq. (7):

$$\min_{\theta \in \Theta} \Big( \max_{\phi \in \Phi} \mathbb{E}_\phi \Big[\sum_{t=1}^{T} r_\theta(\boldsymbol{s}^t_{true}, a^t)\Big] - \frac{1}{\eta}R(\phi) \Big) - \sum_{t=1}^{T} r_\theta(\boldsymbol{s}^t_{true}, a^t_{true})$$

We make a assumption that there is no repeated pair $(\boldsymbol{s}^t_{true}, a^t)$ in Eq. (7). This is a very soft assumption because $\boldsymbol{s}^t_{true}$ is updated overtime, and $a^t$ is in fact representing its feature vector $\boldsymbol{f}^t_{a^t}$, which is in space $\mathbb{R}^d$. With this assumption, we can let $\phi$ map each pair $(\boldsymbol{s}^t_{true}, a^t)$ to the optimal vector $\phi^{t*}$ which maximize $r_\theta(\boldsymbol{s}^t_{true}, a^t) - \frac{1}{\eta}R(\phi^t)$ since there is no repeated pair. Using Eq. (16), we have

$$\max_{\phi \in \Phi} \mathbb{E}_\phi \Big[\sum_{t=1}^{T} r_\theta(\boldsymbol{s}^t_{true}, a^t)\Big] - \frac{1}{\eta}R(\phi) = \max_{\phi \in \Phi}\sum_{t=1}^{T} \mathbb{E}_\phi \big[r_\theta(\boldsymbol{s}^t_{true}, a^t)\big] - \frac{1}{\eta}R(\phi)$$

$$= \sum_{t=1}^{T} \Big(\sum_{i=1}^{k} \phi_i^{t*} r(\boldsymbol{s}^t, a_i) - \frac{1}{\eta}\sum_{i=1}^{k} \phi_i^{t*}\log\phi_i^{t*}\Big) = \sum_{t=1}^{T} \frac{1}{\eta}\log\Big(\sum_{i=1}^{k} \exp(\eta r_\theta(\boldsymbol{s}^t_{true}, a_i))\Big).$$

Eq. (7) can then be written as

$$\min_{\theta \in \Theta}\sum_{t=1}^{T} \frac{1}{\eta}\log\Big(\sum_{i=1}^{k} \exp(\eta r_\theta(\boldsymbol{s}^t_{true}, a_i))\Big) - \sum_{t=1}^{T} r_\theta(\boldsymbol{s}^t_{true}, a^t_{true}),$$

which is the negative log-likelihood function and is equivalent to lemma 2. $\square$

# B  ALGORITHM BOX

The following is the algorithm of learning the cascading deep Q-networks. We employ the cascading $Q$ functions to search the optimal action efficiently (line 9). Besides, both the experience replay (Mnih et al., 2013) and $\varepsilon$-exploration techniques are applied. The system's experiences at each time-step are stored in a replay memory set $\mathcal{M}$ (line 11) and then a minibatch of data will be sampled from the replay memory to update $\widehat{Q^j}$ (line 13 and 14). An exploration to the action space is executed with probability $\varepsilon$ (line 8).

---

**Algorithm 2** cascading deep Q-learning (CDQN) with Experience Replay

---

1: Initialize replay memory $\mathcal{M}$ to capacity $N$
2: Initialize parameter $\Theta_j$ of $\widehat{Q^j}$ with random weights for each $1 \leq j \leq k$
3: **for** iteration $i = 1$ to $L$ **do**
4:     Sample a batch of users $\mathcal{U}$ from training set
5:     Initialize the states $s^0$ to a zero vector for each $u \in \mathcal{U}$
6:     **for** $t = 1$ to $T$ **do**
7:         **for** each user $u \in \mathcal{U}$ simultaneously **do**
8:             With probability $\varepsilon$ select a random subset $\mathcal{A}^t$ of size $k$
9:             Otherwise, $\mathcal{A}^t = \text{ARGMAX\_Q}(s_u^t, \mathcal{I}^t, \Theta_1, \cdots, \Theta_k)$
10:            Recommend $\mathcal{A}^t$ to user $u$, observe user action $a^t \sim \phi(s^t, \mathcal{A}^t)$ and update user state $s^{t+1}$
11:            Add tuple $\left(s^t, \mathcal{A}^t, r(s^t, a^t), s^{t+1}\right)$ to $\mathcal{M}$
12:        **end for**
13:        Sample random minibatch $B \overset{\text{iid.}}{\sim} \mathcal{M}$
14:        For each $j$, update $\Theta_j$ by SGD over the loss $\left(y - \widehat{Q^j}(s^t, A_{1:j}^t; \Theta_j)\right)^2$ for $B$
15:    **end for**
16: **end for**
17: **return** $\Theta_1, \cdots, \Theta_k$

---

# C  DATASET DESCRIPTION

**(1) MovieLens public dataset**[1] contains large amounts of movie ratings collected from their website. We randomly sample 1,000 active users from this dataset. On average, each of these active users rated more than 500 movies (including short films), so we assume they rated almost every movie that they watched and thus equate their rating behavior with watching behavior. MovieLens dataset is the most suitable public dataset for our experiments, but it is still not perfect. In fact, none of the public datasets provides the context in which a user's choice is made. Thus, we simulate this missing information in a reasonable way. For each movie watched(rated) on the date $d$, we collect a list of movies released within a month before that day $d$. On average, movies run for about four weeks in theater. Even though we don't know the actual context of user's choice, at least the user decided to watch the rated movie instead of other movies in theater. Besides, we control the maximal size of each displayed set by 40. **Features:** In MovieLens dataset, only titles and IDs of the movies are given, so we collect detailed movie information from Internet Movie Database(IMDB). Categorical features as encoded as sparse vectors and descriptive features are encoded as dense vectors. The combination of such two types of vectors produces 722 dimensional raw feature vectors. To further reduce dimensionality, we use logistic regression to fit a wide&deep networks (Cheng et al., 2016) and use the learned input and hidden layers to reduce the feature to 10 dimension.

**(2) An online news article recommendation dataset from Ant Financial** is anonymously collected from Ant Financial news article online platform. It consists of 50,000 users' clicks and impression logs for one month, involving dozens of thousands of news. It is a time-stamped dataset which contains user features, news article features and the context where the user clicks the articles. The size of the display set is not fixed, since a user can browse the news article platform as she likes.

---

[1]https://grouplens.org/datasets/movielens/

On average a display set contains 5 new articles, but it actually various from 2 to 10. **Features:** The news article raw features are approximately of dimension 100 million because it summarizes the key words in the article. Apparently it is too expensive to use these raw features in practice. The features we use in the experiments are 20 dimensional dense vector embedding produced from the raw feature by wide&deep networks. The reduced 20 dimensional features are widely used in this online platform and revealed to be effective in practice.

**(3) Last.fm**[2] contains listening records from 359,347 users. Each display set is simulated by collecting 9 songs with nearest time-stamp.

**(4) Yelp**[3] contains users' reviews to various businesses. Each display set is simulated by collecting 9 businesses with nearest location.

**(5) RecSys15**[4] contains click-streams that sometimes end with purchase events.

**(6) Taobao**[5] contains the clicking behavior and buying behavior of users in 22 days. We consider the buying behaviors as positive events.

## D MORE FIGURES FOR EXPERIMENTAL RESULTS

### D.1 FIGURES FOR SECTION 6.2

An interesting comparison is shown in Figure 4 and more similar figures are provided here. The blue curve is the trajectory of a user's actual choices of movies over time. The orange curves are simulated trajectories predicted by GAN and CCF, respectively. Similar to what we conclude in section 6.2, these figures reveal the good performances of GAN user model in terms of capturing the evolution of users' interest.

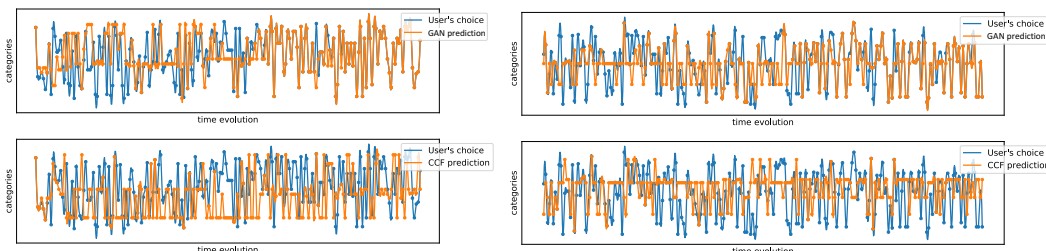

Figure 8: Two more examples: comparison of the true trajectory(blue) of user's choices, the simulated trajectory predicted by GAN model (orange curve in upper sub-figure) and the simulated trajectory predicted by CCF (orange curve in the lower sub-figure) for the same user. $Y$-axis represents 80 categories of movies.

### D.2 FIGURES FOR SECTION 6.3

We demonstrate the policy performance in user level in figure 5 by comparing the cumulative reward. Here we attach the figure which compares the click rate. In each sub-figure, red curve represents GAN-DQN policy and blue curve represents the other. GAN-DQN policy contributes higher averaged click rate for most users.

### D.3 FIGURES FOR SECTION 6.4

This figure shows three sets of results corresponding to different sizes of display set. It reveals how users' cumulative reward(averaged over 1,000 users) increases as each policy interacts with and adapts to 1,000 users over time. It can be easily that the CDQN policy pre-trained over a GAN user model can adapt to online users much faster then other model-free policies and can reduce the risk

---

[2]https://www.last.fm/api
[3]https://www.yelp.com/dataset/
[4]https://2015.recsyschallenge.com/
[5]https://tianchi.aliyun.com/datalab

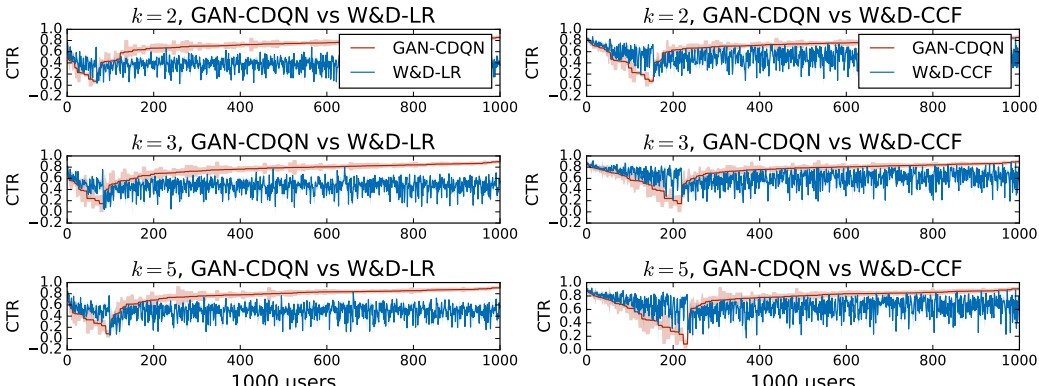

Figure 9: Comparison of click rates among 1,000 users under the recommendation policies based on different user models. In each figure, red curve represents GAN-DQN policy and blue curve represents the other. The experiments are repeated for 50 times and standard deviation is plotted as the shaded area. This figure is similar to figure 5, except that it plots the value of click rates instead of user's cumulative rewards.

of losing the user at the beginning. The experiment setting is similar to section 6.3. All policies are evaluated on a separated set of 1,000 users associated with a test model. We need to emphasize that the GAN model which assists the CDQN policy is learned from a training set of users without overlapping test users. It is different from the test model which fits the 1,000 test users.

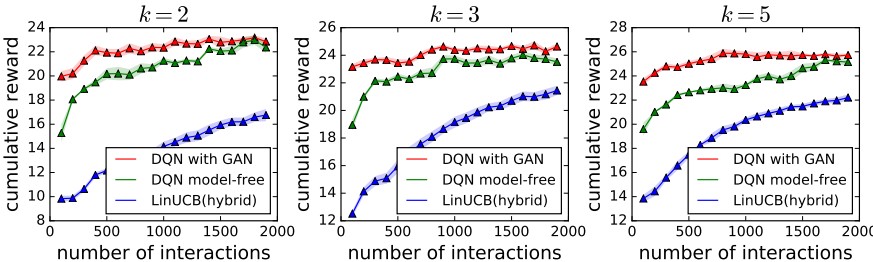

Figure 10: Comparison of the averaged cumulative reward among 1,000 users under different adaptive recommendation policies. $X$-axis represents how many times the recommender interacts with online users. Here the recommender interact with 1,000 users each time, so in fact each interaction represents 100 online data points. $Y$-axis is the click rate. Each point $(x, y)$ in this figure means a click rate $y$ is achieved after $x$ many times of interactions with the users.

