# OpenReview forum: "Neural Model-Based Reinforcement Learning for Recommendation"
_ICLR.cc/2019/Conference_

### Official Review · AnonReviewer1 · 2018-11-03
**Review for Neural Model-Based Reinforcement Learning for Recommendation**

**Rating:** 5
**Confidence:** 5

**Review:**

The authors propose a deep reinforcement learning based recommendation algorithm. Instead of manually designing reward function for RL, a generative adversarial network was proposed to learn the reward function based on user's dynamic behavior. The authors also try to provide an efficient combinatorial recommendation algorithm by designing a cascade DQN. The authors hold their experiments on the Movielens and Ant Financial news dataset. The authors adopt logistic regression (LR) and collaborative competitive filtering(CCF) as comparison baseline to evaluate recommendation performance. The authors also compared their proposed RL policy CQDN with LinUCB.

[Pros in Summary]
1. Recommendation in the deep neural network based RL is a hot topic.

2. The motivation for using a self-learned rewards function and provide efficient combinatorial recommendation is interesting.

[Cons in Summary]
1. The motivations/claimed contributions are not well supported/illustrated by the proposed algorithm or experiments.

2. Some assumptions may not be realistic.

3. The experiment is not sufficient without enough state of art baselines.

4. The writing of this paper needs improvement.

[Thoughts, Questions, and Problems in Details]
1. The idea of using a learned reward function instead of manually defined one sound sweet. But based on (7) and (8), the reward function is essentially giving more rewards for the action that the user really clicks on. How much difference is there compared with traditional manual reward design of giving a click with a reward of 1, especially given the circumstance that a lot of manual intervention is actually used in designing loss function like (7)?

Moreover, in the experiment, there is no comparison experiment evaluating the difference between using a self-learned reward function vs. a traditional manual designed reward function.

2. The assumption "in each pageview, users are recommended a page of k items and provide feedback by clicking on only one of these items; and then the system recommends a new page of k items" does not sound realistic. What if the users click on multiple items?

3. The combinatorial recommendation is useful in the recommendation setting. But it is also important to get the correct ranking order for items from the recommendation list, ie, the best item should rank on the top of the list. Is this principal guaranteed in the combinatorial recommendation proposed in this paper? It is not discussed in this paper.

4. The authors claim to provide an efficient combinatorial recommendation but fail to provide any computational complexity analysis or providing any analysis on training or serving time. Is the proposed algorithm computationally practical to be deployed in a real system?

5. The experiments are too weak because the baselines are old and state of art methods are missing from the comparison.

6. Typos and grammar errors across the paper, to name a few

"we will also estimate a user behavior model associate with the reward function"

"a model for the sequence of user clicking behavior, discussion its parametrization and parameter estimation."

---

> ### Author Response · Authors · 2018-11-25
> **Response to Reviewer1**
>
> We appreciate your constructive and detailed comments! We present our clarification in the following:
>
> (1)The idea of using a learned reward function instead of manually defined one sound sweet. But based on (7) and (8), the reward function is essentially giving more rewards for the action that the user really clicks on.
>
> One can interpret the mini-max framework in two ways:
> (i)The user behavior model \phi acts as a generator which generates the user's next actions based on her history, while the reward r acts as a discriminator which tries to differentiate user's actual actions from those generated by the behavior model \phi. More specifically, the learned reward function r will extract some statistics from both real user actions and model user actions, and try to magnify their differences (or make a larger negative gap). In contrast, the learned user behavior model will try to make the difference smaller, and hence more similar to the real user behavior.
> (ii)Alternatively, the optimization can also be interpreted as a game between an adversary and a learner where the adversary tries to minimize the reward of the learner by adjusting r, while the learner tries to maximize its reward by adjusting \phi to counteract the adversarial moves.  This gives the user behavior training process a large-margin training flavor, where we want to learn the best model even for the worst scenario.
>
> We added these additional intuitive descriptions in sec 4.1 and 4.3 to make it more clear.
>
> (2)How much difference is there compared with traditional manual reward design of giving a click with a reward of 1?
>
> First, the learned reward function provides more information about a user’s preference, and provide a better interpretation of user behavior. Setting reward function to 1/-1 cannot fully differentiate user’s preference over different items.
>
> Second, the learned reward function also helps reinforcement learning to learn a better policy. This can be explained by the reward shaping phenomenon in reinforcement learning, where continuous reward signals can help reinforcement learning algorithm converge better than sparse binary signals. We’ve also included an experimental comparison with the reward as 1/-1 and showed that the learned continuous rewards lead to better policies.
>
> (3)The assumption "in each pageview, users are recommended a page of k items and provide feedback by clicking on only one of these items; and then the system recommends a new page of k items" does not sound realistic. What if the users click on multiple items?
>
> We model the multiple-click case as a sequence of clicks with the same display-set. Essentially, we need an ordered list to fit either our position weighting scheme or LSTM.
>
> (4)The combinatorial recommendation is useful in the recommendation setting. But it is also important to get the correct ranking order for items from the recommendation list, ie, the best item should rank on the top of the list. Is this principal guaranteed in the combinatorial recommendation proposed in this paper? It is not discussed in this paper.
>
> We can use the user behavior model \phi together with the cascading Q-networks to address the ranking question:
> (i) First, the cascading network will select k items from the candidate pool.
> (ii) Then, the user model can be used to assign a likelihood to each selected item. The item with high likelihood will be ranked higher.
> We note that the cascading Q-networks themselves do not explicitly guarantee the ranking of individual items, and the networks will score a set of items jointly. We can use the user behavior model to rank because experiments in Sec 6.2 already show that our user behavior model performs well in terms of ranking the displayed items.
>
> (5)The authors claim to provide an efficient combinatorial recommendation but fail to provide any computational complexity analysis or providing any analysis on training or serving time. Is the proposed algorithm computationally practical to be deployed in a real system?
>
> First, to search the optimal action, there are (n choose k) = n! / (k!(n-k)!) many candidates. With our designed cascaded Q-networks, we only need to search over n candidates for k times. Thus, we can obtain the optimal action with O(kn) computations. We mention this briefly in the last paragraph in sec 5.1.
>
> (6)The experiments are too weak because the baselines are old and state of art methods are missing from the comparison.
>
> In the revised version, we’ve compared to 7 strong baselines in 6 datasets. Besides, we want to clarify that the previous 2 baseline methods(LR and CCF) are already strong baselines since they’ve been augmented with wide & deep feature layers.

---

### Official Review · AnonReviewer2 · 2018-11-03

**Rating:** 6
**Confidence:** 3

**Review:**

This paper belongs to the space of treating recommendation as a reinforcement learning problem, and proposed a model-based (cascaded DQN) approach, using a generative adversarial network to simulate user rewards.

Pros:
+ proposed a set of cascading Q functions, to learn a recommendation policy
+ unified min-max optimization to learn the behavior model and the reward function
+ interesting idea of using generative adversarial networks to simulate user rewards.

Cons:
- in Figure 6 no comparison with model-free (policy-gradient type) of approaches
- there is not a lot of detail on the value of the generative adversarial network for the user behavior dynamics, thus this prevents the reader from fully understanding the contribution
- only 2 datasets are used
- only 100 users for test users seems few
- why only 1000 active users were sampled from MovieLens?

Personally, I would prefer less details on formulating the recommendation problem as an RL problem (as there have been other papers before with a similar formulation) and more detail  on the simulation user reward model, and in general in sections 4 and 5. Also, the experiments could be strengthened.

---

> ### Author Response · Authors · 2018-11-25
> **Response to Reviewer2**
>
> Thank you very much for your review and suggestions! We present our clarification in the following:
>
> (1)in Figure 6 no comparison with model-free (policy-gradient type) of approaches
>
> In Figure 6 (now Figure 7 in revised version), we compared deep Q-learning without using user model for adaption. Furthermore, LinUCB is another model-free approach which assumes an adversarial user. In both cases, our model-based adaptation produces better results.
>
> (2)there is not a lot of detail on the value of the generative adversarial network for the user behavior dynamics, thus this prevents the reader from fully understanding the contribution
>
> Our GAN framework learns both users model and the corresponding reward function in a unified framework. The values are reflected in:
> (i) The framework allows us to learn a better user model by using the learned the loss function (the reward r).
> (ii) The framework allows later reinforcement learning to be carried out with a principled reward function, rather than manually designed reward.
> (iii) The framework allows us to perform model-based RL and online adaptation for new users to achieve better results.
>
> (3)only 2 datasets are used
>
> In the revised version, we compared with 7 strong baselines in 6 datasets. In most datasets, our method achieves the best results.
>
> (4)only 100 users for test users seems few
>
> We have the policies tested on 1,000 users, but in the first version, we only plotted the results on 100 users. In the revised version, we updated the figures and the numbers to present the results on 1,000 users.

---

### Official Review · AnonReviewer3 · 2018-11-04
**Interesting problem and ideas, manuscript may not be ready for publication yet**

**Rating:** 5
**Confidence:** 4

**Review:**

This paper proposes to frame the recommendation problem as one of (model-based) RL. The two main innovations are: 1) a model and objective for learning the environment and reward models; 2) a cascaded DQN framework for reasoning about a combinatorial number of actions (i.e., which subset of items to recommend to the user).

The problem is clearly important and the authors' approach focuses on solving some of the current issue with deployment of RL-based recommenders. Overall the paper is relatively easy to follow, but the current version is not the easiest to understand and, in particular, it may be worth providing more intuitions (e.g., about the GAN-like setup). I also found that several decisions are not properly justified. The novelty of this paper seems reasonably high but my impression is that other/stronger baselines would make the study more convincing. Copy-editing the paper would also greatly improve readability.

Detailed comments:
- I am not clear on whether or not in the proposed model, users are "allowed" to not click on a recommendation. It sounds like the authors in fact allow it but I think that could be made clearer.

- Section 4. I am not sure that using the Generative Adversarial Network terminology is useful here. Specifically, it is not clear what is your generative model over (I imagine next state and reward?).

- Remark in Section 4.1: It seems like a user not clicking on something is also useful information. Why not model it?

- I am a bit unclear on the value of Lemma 1. Further, what are the assumptions behind it? (also what is this temperature parameter eta?)

- In Section 4.2, the size of your model seems to grow linearly with the number of user interactions. That seems like a major advantage of RNNs/LSTMs. In practice, I imagine you cull the history at some fixed point?

- What is the advantage of learning a reward? E.g., a very simple reward would be to give a positive reward if a user clicks on a recommended item and a negative reward otherwise. What does your learned reward allow beyond this?

- Section 4.3. I also found Section 4.3 to be relatively unclear. I find that more intuition would be helpful.

  Also, if Eq. 7 is equivalent to Eq. 8, then why is the solution of 8 used only to initialize 7? I guess it may have to do with not finding the global optimum.

- Your cascading DQN idea seems like a good one. It would be nice to check if the constraints are correctly learned. If not, this seems like it would do not better than a greedy action-by-action solution. Is that correct?

- In Section 6.1, it would be good to discuss the pre-processing in the main text since it's pretty important to understand the study (e.g., evaluate is impact).

- In 6.2, your baselines seem a bit weak. Why not compare to more recent CF models (e.g., including Session-Based RNNs which you cite earlier)?

- Related work: it would probably be good to survey some of the multi-arm bandit literature. There is also some CF-RL work which should be cited (perhaps there are a few things in there that should be compared to in Section 6.3 & 6.4).

- Section 6.2 and Table 1. I believe that Recall@k is most common in recommendation-systems-for-implicit-data literature. Or, are you assuming that what people do not click on are true negatives? This doesn't seem quite right as users are only allowed to click on a single item.

- In Section 6.3, could you clarify how do you learn your reward model that is used to train the various methods?

- There are many typos and grammatical errors in the paper. I would suggest that the authors carefully copy-edit the manuscript.

---

> ### Author Response · Authors · 2018-11-25
> **Response to Reviewer3**
>
> Thanks for your effort in providing this detailed review!  We present our clarification in the following:
>
> (1)I am not clear on whether or not in the proposed model, users are "allowed" to not click on a recommendation.
>
> ‘Not click’ is always treated as one action in each pageview. In the revised paper (Sec 4.1 Remark(ii)), we provide more explanation to make it clearer.
>
> (2)Section 4. I am not sure that using the Generative Adversarial Network terminology is useful here.
>
> We want to generate the user’s next action based on her current state (i.e. her historical sequence of actions). In other words, the behavior model \phi aims at mimicking the user’s behavior as a result of optimizing an unknown reward function r. Thus, one needs to simultaneously estimate \phi and r. The estimation framework is a mini-max optimization resembling the generator(\phi) and discriminator(r). The benefit of this GAN framework is that one can view the learning of the reward r as learning a loss function for the generative model \phi. The learned loss function can lead to a better user behavior model than obtained via a predefined loss function.
>
> In the revised paper, we enriched the description and explanation of both the user model and its mini-max formulation in Sec 4.1 and 4.3.
>
> (3)Remark in Section 4.1: It seems like a user not clicking on something is also useful information. Why not model it?
>
> ‘No click’ is always modeled as one action. We do not use a specific notation to indicate ‘no click’. Instead, it is denoted as one of the items. If the user does not click, then she is clicking the ‘no click’ item. In the revised paper (Sec 4.1 Remark(ii)), we’ve clarified this.
>
> (4)I am a bit unclear on the value of Lemma 1. Further, what are the assumptions behind it? (also what is this temperature parameter eta?)
>
> Lemma 1 has two major values:
> (i) It helps the model interpretation and makes the exploration-exploitation nature of the model become clear. Our general formulation can use many different regularization terms, such as L2 regularization and f-divergence, beyond just Shannon entropy function, to induce potentially different user behaviors. This is analogous to generative adversarial networks where a different variational form of the divergence can be used, such as Jensen-Shannon divergence, f-divergence, and Wasserstein divergence. Lemma 1 holds only when the regularization function is the negative Shannon entropy. For other regularization functions, the resulting user behavior model does not have a closed form, but also induces some form of exploration-exploitation trade-off.
> (ii) Lemma 1 is also used to prove Lemma 2, which gives us a way to initialize the mini-max optimization problems and make the training more stable for more general divergences.
>
> From equation (3), the regularization parameter eta represents the exploration level of the user.  When eta is smaller, the user is more exploratory. When eta is larger, the user is more stubborn to choose the item with the highest reward. We discuss the parameter eta under Lemma 1 in the revised paper.
>
> (5)In Section 4.2, the size of your model seems to grow linearly with the number of user interactions. That seems like a major advantage of RNNs/LSTMs. In practice, I imagine you cull the history at some fixed point?
>
> Yes, we use a fixed time window of history for the position weigh model. In practice, Backpropagation over time in RNN/LSTM also stops at certain fixed time steps.
>
> (6)What is the advantage of learning a reward?
>
> First, the learned reward function provides more information about a user’s preference, and provide a better interpretation of user behavior. Setting reward function to 1/-1 cannot fully differentiate user’s preference over different items.
>
> Second, the learned reward function also helps reinforcement learning to learn a better policy. This can be explained by the reward shaping phenomenon in reinforcement learning, where continuous reward signals can help reinforcement learning algorithm converge better than sparse binary signals. We’ve also included an experimental comparison with the reward as 1/-1 and showed that the learned continuous rewards lead to better policies in the revised version.

---

> > ### Author Response · Authors · 2018-11-25
> > **Response to Reviewer3(continue)**
> >
> > (7)Section 4.3 to be relatively unclear.
> >
> > One can interpret the mini-max optimization in two ways:
> > The user behavior model \phi acts as a generator which generates the user's next actions based on her history, while the reward r acts as a discriminator which tries to differentiate user's actual actions from those generated by the behavior model \phi. More specifically, the learned reward function r will extract some statistics from both real user actions and model user actions, and try to magnify their differences (or make a larger negative gap). In contrast, the learned user behavior model will try to make the difference smaller, and hence more similar to the real user behavior.
> > Alternatively, the optimization can also be interpreted as a game between an adversary and a learner where the adversary tries to minimize the reward of the learner by adjusting r, while the learner tries to maximize its reward by adjusting \phi to counteract the adversarial moves.  This gives the user behavior training process a large-margin training flavor, where we want to learn the best model even for the worst scenario.
> >
> > We added these additional intuitive descriptions in sec 4.1 and 4.3 to make it more clear.
> >
> > (8)if Eq. 7 is equivalent to Eq. 8, why is the solution of 8 used only to initialize 7?
> >
> > The general formulation in Eq7 can use many different regularization terms, such as L2 regularization and f-divergence, beyond just Shannon entropy function, to induce potentially different user behaviors. This is analogous to GAN where a different variational form of the divergence can be used, such as Jensen-Shannon divergence, f-divergence, and Wasserstein divergence.
> >
> > Eq9 in the revised version (previous Eq8) has two use cases:
> > (i) If Shannon entropy is used, we can directly use it for learning reward function and the user behavior model is related to the reward in closed form.
> > (ii) If other regularizations are used, the reward function from Eq9 can be used as initialization for the general optimization algorithm in Eq8 which works for any regularization. We include the experiments of training our model with L2 regularization using this initialization scheme in the revised paper.
> >
> > (9)Your cascading DQN idea seems like a good one. It would be nice to check if the constraints are correctly learned. If not, this seems like it would do no better than a greedy action-by-action solution.
> >
> > Thank you for suggesting the constraint-checking! We empirically plot the quantity in the left-hand side(as y) and the right-hand side(as x) of the constraints. The scatter-plot is included as Figure 6, where we observe that the points are approximately along the diagonal.
> >
> > Also, to make an action-by-action greedy policy, we design another action value function as Q(s, a1, a2, a3)=Q(s, a1)+Q(s, a2)+Q(s, a3) and include the comparison of its performance in table 2. Our cascading Q-network is much better than this greedy approach. Especially, when k is larger, the gap between their performances is larger.
> >
> > (10)In Section 6.1, it would be good to discuss the pre-processing in the main text.
> >
> > In the revised version, we added some descriptions to the main text.
> >
> > (11)In 6.2, your baselines seem a bit weak.
> >
> > In the revised version, we’ve compared to 7 strong baselines in 6 datasets. Besides, we want to clarify that the previous 2 baseline methods(LR and CCF) are already strong baselines since they’ve been augmented with wide & deep feature layers.
> >
> > (12)Related work: it would probably be good to survey some of the multi-arm bandit literature. There is also some CF-RL work which should be cited.
> >
> > Previous RL-based methods are either using manually designed reward (eg. +1/-1 for click/no click) or model free. In the revision, we’ve compared with RL methods with manually designed rewards and model-free based RL approach, as well as multi-arm bandit based method (LinUCB). Our method is consistently better than these alternatives.
> >
> > (13)Section 6.2 and Table 1. I believe that Recall@k is most common in recommendation-systems-for-implicit-data literature. Or, are you assuming that what people do not click on are true negatives? This doesn't seem quite right as users are only allowed to click on a single item.
> >
> > The experiments In Section 6.2 is to show that our user behavior model can make the most accurate prediction for users' behavior. Essentially, given user’s previous choices, we want to predict what is her *next choice*. In this case, only 1 item is positive, so we only report precision.
> >
> > (14)In Section 6.3, could you clarify how do you learn your reward model that is used to train the various methods?
> >
> > The reward model learned together with user’s behavior model via the mini-max optimization. Once both are learned, it is treated as the simulation environment needed for reinforcement learning. Then various RL policies can be learned by interacting with this environment.

---

### Author Response · Authors · 2018-11-25
**Paper Revision 1**

Dear reviewers,

Thank you for your insightful suggestions! We have revised our paper to address some common concerns as well as individual questions. As for common questions,

(1)  We’ve compared with 7 strong baselines in 6 different datasets.
Besides, we want to clarify that the previous 2 baseline methods (LR and CCF)  are actually strong baselines since both have wide & deep(W&D) feature layers.

(2) We’ve clarified the explanation of the GAN user model and the mini-max formulation.
We reduced the length of introducing the RL framework and added more interpretation for the user model as well as the mini-max formulation in sec 4.1 and 4.3.

(3) We’ve improved the writing.
Typos and grammar errors are carefully adjusted.

As for reviewer’s individual questions, we will clarify separately below.

---

### Meta-Review · Area_Chair1 · 2018-12-13
**Improvements needed**

**Confidence:** 5
**Recommendation:** Reject

**Metareview:**

This paper formulates the recommendation as a model-based reinforcement learning problem. Major concerns of the paper include: paper writing needs improvement; many decisions in experimental design were not justified; lack of sufficient baselines; results not convincing. Overall, this paper cannot be published in its current form.